# Identification of antibiotics for use in selection of the chytrid fungi *Batrachochytrium dendrobatidis* and *Batrachochytrium salamandrivorans*

Kristyn A. Robinson, Mallory Dunn, Shane P. Hussey, Lillian K. Fritz-Laylin⬤ *

Department of Biology, The University of Massachusetts Amherst, Amherst, MA, United States of America

* lfritzlaylin@umass.edu

## Abstract

Global amphibian populations are being decimated by chytridiomycosis, a deadly skin infection caused by the fungal pathogens *Batrachochytrium dendrobatidis* (*Bd*) and *B. salamandrivorans* (*Bsal*). Although ongoing efforts are attempting to limit the spread of these infections, targeted treatments are necessary to manage the disease. Currently, no tools for genetic manipulation are available to identify and test specific drug targets in these fungi. To facilitate the development of genetic tools in *Bd* and *Bsal*, we have tested five commonly used antibiotics with available resistance genes: Hygromycin, Blasticidin, Puromycin, Zeocin, and Neomycin. We have identified effective concentrations of each for selection in both liquid culture and on solid media. These concentrations are within the range of concentrations used for selecting genetically modified cells from a variety of other eukaryotic species.

## Introduction

Chytrids are early diverging fungi that are commonly found in aquatic and moist environments [1]. They play key ecological roles, particularly by cycling carbon between trophic levels [2, 3]. Chytrids have a biphasic life cycle characterized by motile and sessile stages (**Fig 1**) [4–6]. They begin their life as motile "zoospores," which use a flagellum to swim through water and, for some species, actin-based motility to crawl along surfaces [7, 8]. Zoospores then transition to a sessile growth stage by retracting their flagellum and building a cell wall in a process referred to as encystation. Encysted spores of many species develop into sporangia and develop hyphal-like structures called rhizoids and grow rapidly. Each sporangium produces many zoospores that exit via discharge papillae to begin the life cycle anew.

Many chytrids are pathogens that infect protists, plants, algae, fungi, and vertebrates [2]. The most infamous chytrids are the vertebrate pathogens *Batrachochytrium dendrobatidis* (*Bd*) and *B. salamandrivorans* (*Bsal*). Both pathogens cause chytridiomycosis, a skin disease plaguing amphibians worldwide [4, 6]. Recent estimates indicate that *Bd* has affected several hundred amphibian species and has been recorded on every continent except for Antarctica [9–11]. *Bsal* was more recently discovered in 2013 after a steep decline in fire salamander populations in Belgium [6].

**Data Availability Statement:** All relevant data are within the manuscript and its Supporting Information files.

**Funding:** This work was supported by the National Science Foundation (IOS 1827257), awarded to Lillian K Fritz-Laylin (LFL). https://www.nsf.gov/funding/pgm_summ.jsp?pims_id=505480 The funders had no role in study design, data collection and analysis, decision to publish, or preparation of the manuscript.

**Competing interests:** The authors have declared that no competing interests exist.

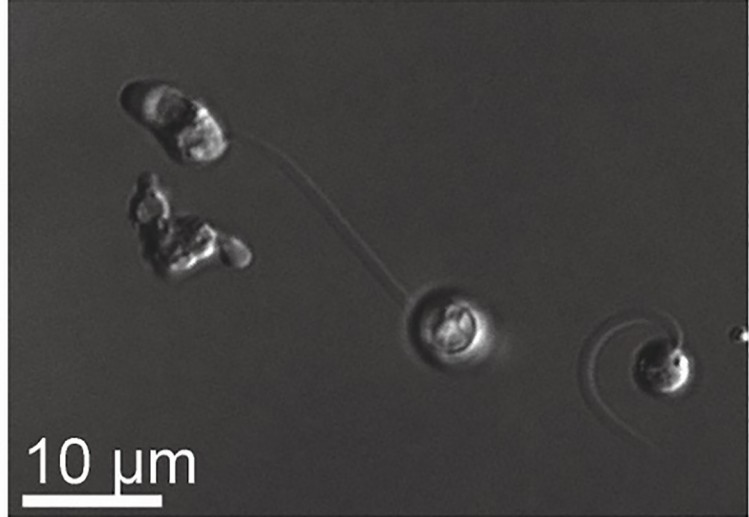

**Fig 1. Life cycle of chytrid fungi.** As illustrated here with images of *Bsal*, chytrid fungi have a biphasic life cycle characterized by a stationary growth phase called a sporangium (top) and a motile dispersal phase called a zoospore (bottom). Images taken at 100X using differential interference contrast (DIC) microscopy.

Management strategies for these pathogens have been developed and implemented in limited contexts, but implementation in real world settings remains a challenge. To develop better treatments, we need to understand the biology of chytrids in order to identify targets for drug development. However, studying the molecular mechanisms driving pathogenesis remains challenging due to the lack of genetic tools available for chytrid fungi. Electroporation protocols have been developed for *Bd* and *Bsal*, which could be used to deliver molecular payloads for genetics manipulation such as plasmids and/or CRISPR-Cas9 complexes [12]. The recent success in genetic manipulation of a related chytrid species, *Spizellomyces punctatus* (*Sp*), is a major breakthrough for our ability to study chytrid biology [8]. We and others are now striving to adapt this technology to *Bd* and *Bsal* to further our understanding of chytridiomycosis.

A key step to genetic tool development is the identification of methods for selection of successful transformants. The most commonly used selection method is antibiotic resistance:

incorporating a gene that provides specific drug resistance allows transformed cells to survive exposure to the antibiotic while all of the other cells are killed [13]. Distinct classes of antibiotics are commonly used for selection, each with their own molecular targets and corresponding organismal specificity. In addition to testing whether a given antibiotic kills cells of interest, it is important to pay attention to the effective concentration of each antibiotic. This is because a low concentration will not apply sufficient selective pressure and a high concentration could produce off-target effects and kill cells indiscriminately [14].

In this paper, we examine five antibiotics used in fungal and animal systems and identify the effective inhibitory concentration(s) necessary to prevent cell growth in liquid and solid media. Hygromycin, Blasticidin, and Puromycin inhibit protein translation in both bacterial and eukaryotic cells. Hygromycin inhibits protein synthesis by binding to the small ribosomal subunit and stabilizing the tRNA in the A site, preventing the progression of translation [15]. Blasticidin inhibits the terminating step of translation while Puromycin causes the ribosome to prematurely detach from mRNA [16, 17]. Although neomycin targets the prokaryotic 30S ribosomal subunit and causes codon misreading and mistranslation, it has been used in eukaryotes because of the similarity between mitochondrial and chloroplast ribosomes and bacterial ribosomes [18]. Zeocin intercalates in the DNA of both bacteria and eukaryotes and introduces double-stranded breaks, ultimately causing cell death [19].

## Results

To establish appropriate selection compounds for use with *Bd* and *Bsal*, we first identified antibiotics commonly used for selection with both mammalian and fungal systems. We chose five compounds (Hygromycin, Blasticidin, Puromycin, Zeocin, and Neomycin) to test based on the mechanism of action of each compound, their proven efficacy for use with both animal and fungal cells, and the availability of resistance genes (**Table 1**). We next tested the ability of these five compounds to inhibit the growth of *Bd* and *Bsal* cells in liquid culture. Although solid agar media is typically used for colony selection in chytrid and other fungi [8, 20, 21], we chose to use liquid culture to identify initial working concentrations because measuring zoospore release in liquid media is rapid and easily quantified.

To measure the effect of each antibiotic on *Bd* and *Bsal* growth, we added a wide range of antibiotic concentrations to cultures of age matched zoospores and allowed them to grow for one full life cycle: three (*Bd*) or four (*Bsal*) days. We then measured the concentration of released zoospores in each culture. Initial concentrations were selected based on known inhibitory concentrations for other organisms (**Table 1**) and spanned many orders of magnitude. Based on these preliminary experiments (not shown), we then identified possible working concentration ranges for each antibiotic in both species and tested intermediate concentrations using three biological replicates separated in time (**Figs 2 and 3**). To enable comparison of zoospore release from replicate experiments conducted on different days, we normalized counts for each replicate to its antibiotic-free control.

We identified antibiotic concentrations that consistently prevented growth in all three biological replicates—the successful concentrations are highlighted in orange in each figure. We found Hygromycin, Zeocin, Blasticidin and Neomycin could inhibit *Bd* growth in liquid culture (**Fig 2**), while all of the tested antibiotics inhibited *Bsal* growth (**Fig 3**). In *Bd*, Hygromycin has the lowest minimum inhibitory concentration (0.1 µg/ml), followed by Blasticidin (1 µg/ml), Zeocin (5 µg/ml), and Neomycin (600 µg/ml). Puromycin did not inhibit growth in *Bd* with the concentrations tested. In *Bsal*, Zeocin prevented growth at 1 µg/ml, followed by Blasticidin (2 µg/ml), Hygromycin (10 µg/ml), Puromycin (50 µg/ml), and Neomycin (250 µg/ml).

**Table 1. Antibiotic concentrations used to select for gene expression in select eukaryotes.** This table lists the key features of the antibiotics used in this study: the drug class, the target, known resistance genes, the current listed price per gram from Millipore Sigma, and the concentrations used in select eukaryotes. Species include representatives from plants (*Arabidopsis thaliana* and *Chlamydomonas reinhardtii*), protozoa (*Trypanosoma brucei*), amoebae (*Dictyostelium discoideum*), fungi (*Aspergillus spp.*, *Schizosaccharomyces pombe*, *Saccharomyces cerevisiae*), and animals (human) in addition to the two species tested in this study. The lowest concentrations of each antibiotic which inhibited growth in liquid and solid media for *Bd* and *Bsal* are listed from our findings in this study. These concentrations were used to calculate the cost per liter of growth media for both *Bd* and *Bsal*.

| | Drug Class | Target | Known Resistance Genes | List price per gram (Millipore&Sigma) | Lowest drug conc. for growth inhibition for Bd | Cost per Liter for Bd | Lowest drug conc. for growth inhibition for Bsal | Cost per Liter for Bsal | Conc. for HeLa cells | Conc. for hESC | Conc. for Fibroblasts | Conc. for Arabidopsis thaliana | Conc. for Dictyostelium discoideum | Conc. for Trypanosoma brucei | Conc. for Chlamydomonas reinhardtii | Conc. for Aspergillus spp | Conc. for S. pombe | Conc. for S. cerevisiae |
|---|---|---|---|---|---|---|---|---|---|---|---|---|---|---|---|---|---|---|
| **Neomycin** | Amino-glycoside | Ribosome [18] | neo‡ | $1.93/g | Liquid: 600 µg/ml Solid: 1 mg/ml | Liquid: $1.16/L. Solid: $1.93/L. | Liquid: 250 µg/ml Solid: N/A | Liquid: $0.48/L. Solid: >$1.93/L. | - | - | - | - | - | - | 300 µg/ml [22] | 200–400 mg/ml [23] | 0.375 g/L [24] | 6.25 mM [25] |
| **Hygromycin** | Atypical Amino-glycoside | Ribosome [15] | hyg, hph | $998/g | Liquid: 1 µg/ml Solid: 0.1 µg/ml | Liquid: $1.00/L. Solid: $0.10/L. | Liquid: 10 µg/ml Solid: 10 µg/ml | Liquid: $9.98/L. Solid: $9.98/L. | 100–200 µg/ml [26,27] | 40 µg/ml [28] | 40 µg/ml [29] | 15–50 µg/ml [30,31] | 25–40 µg/ml [32] | 5–50 µg/ml [33,34] | 1–20 µg/ml [35] | 100 µg/ml [36] | 400 mg/L [37] | 300 µg/ml [38] |
| **Blasticidin** | Nucleoside Antibiotic | Ribosome [16] | bsr, bls, bsd | $6280/g | Liquid: 5 µg/ml Solid: 10 µg/ml | Liquid: $31.25/L. Solid: $62.80 | Liquid: 2 µg/ml Solid: 10 µg/ml | Liquid: $12.56/L. Solid: $62.80/L | 10–20 µg/ml [39,40] | 2.0 µg/ml [41] | 8 µg/ml [29] | 10 µg/ml [42] | 10 µg/ml [43,44] | 2–10 µg/ml [45–47] | - | - | 30 µg/ml [48,49] | 10 mg/ml [50] |
| **Puromycin** | Amino-nucleoside | Ribosome [17] | pac | $5340/g | Liquid: N/A Solid: 100 µg/ml | Liquid: >$1068/L Solid: $534/L | Liquid: 50 µg/ml Solid: N/A | Liquid: $267/L. Solid: >$2670/L. | 1–2 µg/ml [51–53] | 0.5–5 µg/ml [41,54,55] | 2 µg/ml [29] | - | - | 0.1 µg/ml [56] | - | - | - | ~200 uM [57] |
| **Zeocin** | Glyco-peptide Antibiotic | DNA [19] | ble | $177/g (Invivogen) | Liquid: 10 µg/ml Solid: 10 µg/ml | Liquid: $1.77/L. Solid: $1.77/L. | Liquid: 1 µg/ml Solid: 10 µg/ml | Liquid: $0.18/L. Solid: $1.77/L. | 50 µg/ml [58] | 300 µg/ml [59] | 800 µg/ml [29] | 100 µg/ml [60] | 100 mg/L [61] | - | 5–15 µg/ml [62,63] | 100–125 µg/ml [64] | 150 mg/ml [65] | - |

- no references were found.

\* the organism had to be made susceptible for the antibiotic to work.

‡ the *neo* resistance gene is also used for resistance to the drug G418 which was not tested in this study.

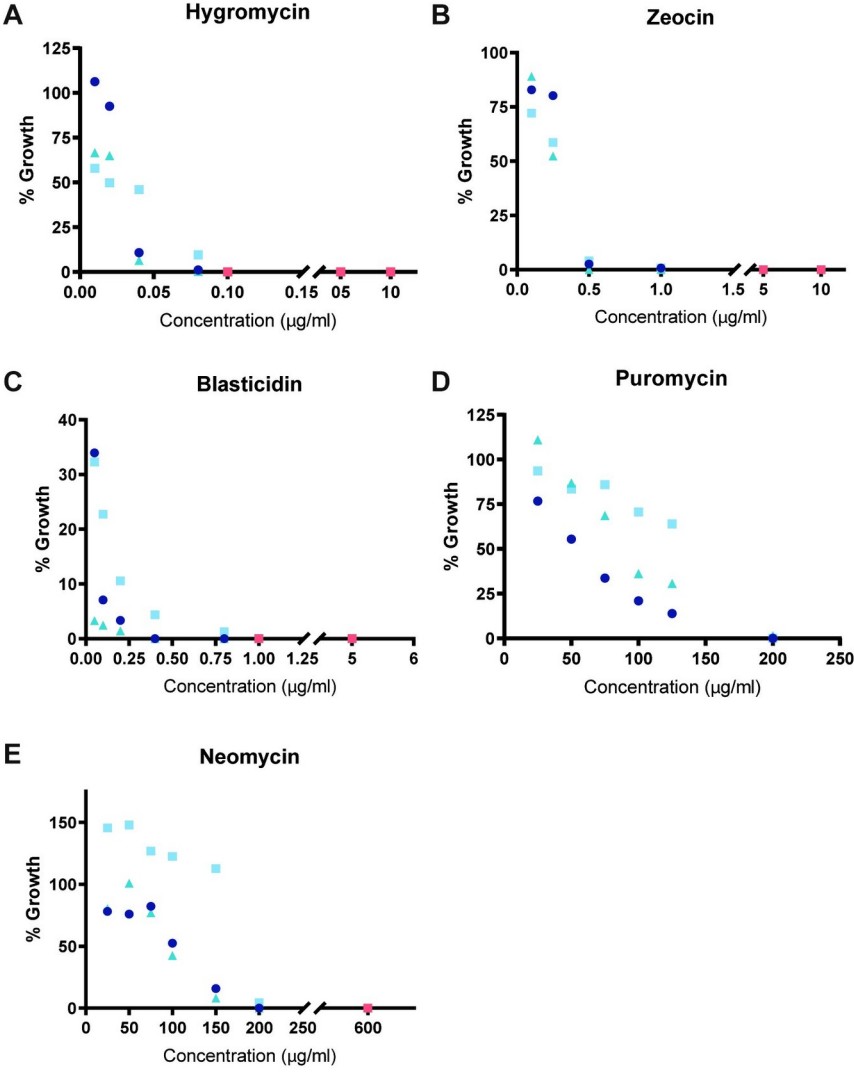

**Fig 2. Inhibition of Bd growth in liquid media.** Percent of *Bd* growth in liquid media supplemented with (A) Hygromycin, (B), Zeocin, (C) Blasticidin, (D) Puromycin, and (E) Neomycin as compared to an antibiotic free control for three temporally isolated replicates (circle, square, and triangle, shades of blue). Orange symbols indicate concentrations at which no growth occurred after three days in all three replicates.

Having identified working concentrations of these compounds for use with liquid media, we next tested their efficacy on solid media. Growing cells on solid media allows for colony formation, which is useful for isolating successful and independent genetic transformants by "picking" colonies that grow under selection. To identify useful concentrations for selection on solid media, we inoculated zoospores on nutrient agar plates containing varying antibiotic concentrations. After a full growth cycle on selective media (three days for *Bd*, four days for *Bsal*), we compared zoospore release to antibiotic-free control cultures by flooding plates with water and looking for motile zoospores (**S1 and S2 Videos**). We defined successful concentrations as those which yielded no zoospore release in either replicate. We found at least one concentration for each antibiotic that prevented zoospore release in the timeframe of a typical growth cycle (**Figs 4 and 5**).

Because detection of colony formation often requires multiple growth cycles, we evaluated the efficiency of growth inhibition by growing plates with no zoospore release for 14 days. We

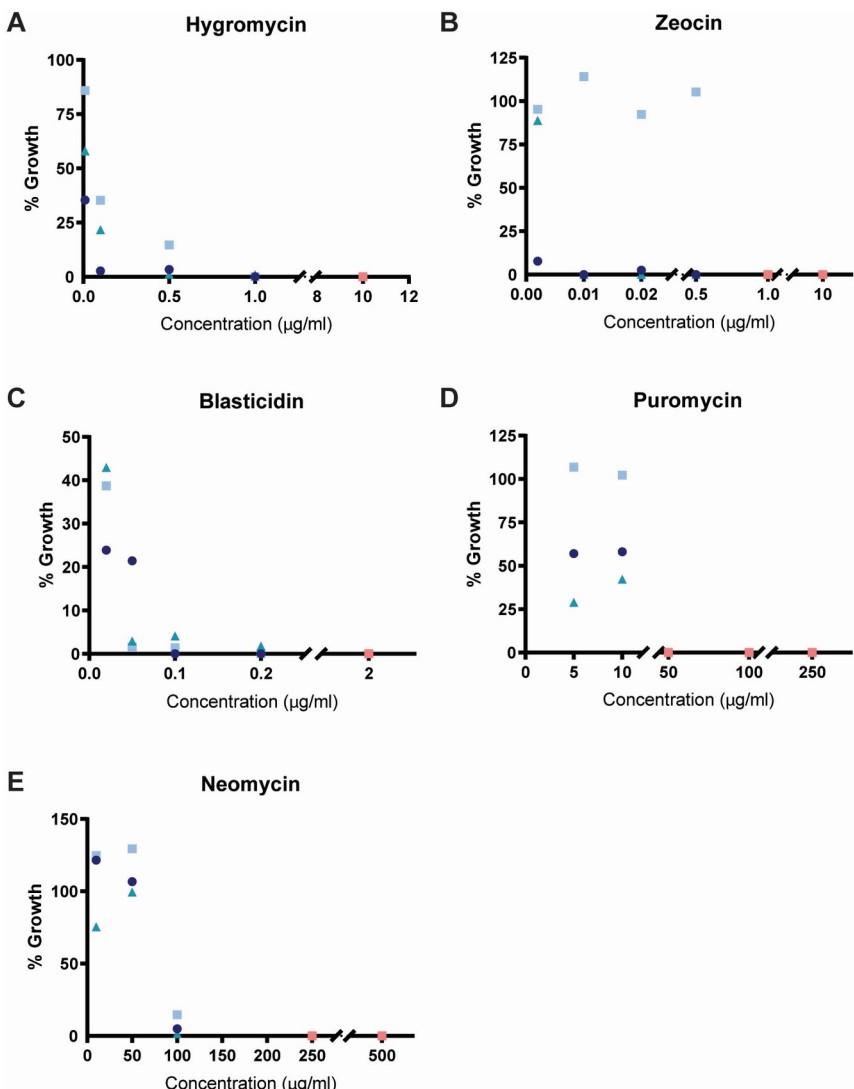

**Fig 3. Inhibition of *Bsal* growth in liquid media.** Percent of *Bsal* growth in liquid media supplemented with (A) Hygromycin, (B), Zeocin, (C) Blasticidin, (D) Puromycin, and (E) Neomycin as compared to an antibiotic free control for three temporally isolated replicates (circle, square, and triangle, shades of blue). Orange symbols indicate concentrations at which no growth occurred after four days in all three replicates.

found that all the tested antibiotics inhibited *Bd* growth on solid media, but only Hygromycin, Blasticidin and Zeocin inhibited growth in *Bsal*. For *Bd*, Hygromycin has the lowest minimum concentration at 0.1 μg/ml, with Blasticidin and Zeocin both following at 10 μg/ml, Puromycin at 100 μg/ml, and Neomycin at 1 mg/ml (**Fig 4**). In *Bsal*, Hygromycin, Blasticidin, and Zeocin all prevented growth for at least 14 days at a concentration of 10 μg/ml, while Puromycin and Neomycin did not prevent growth on solid media (**Fig 5**). The recommended concentrations for selection are highlighted in orange on the tables in both figures (**Figs 4B and 5B**).

## Discussion

This study identified drug concentrations that reproducibly inhibited *Bd* and *Bsal* growth in either liquid culture or on solid media. When a drug worked in both liquid culture and solid

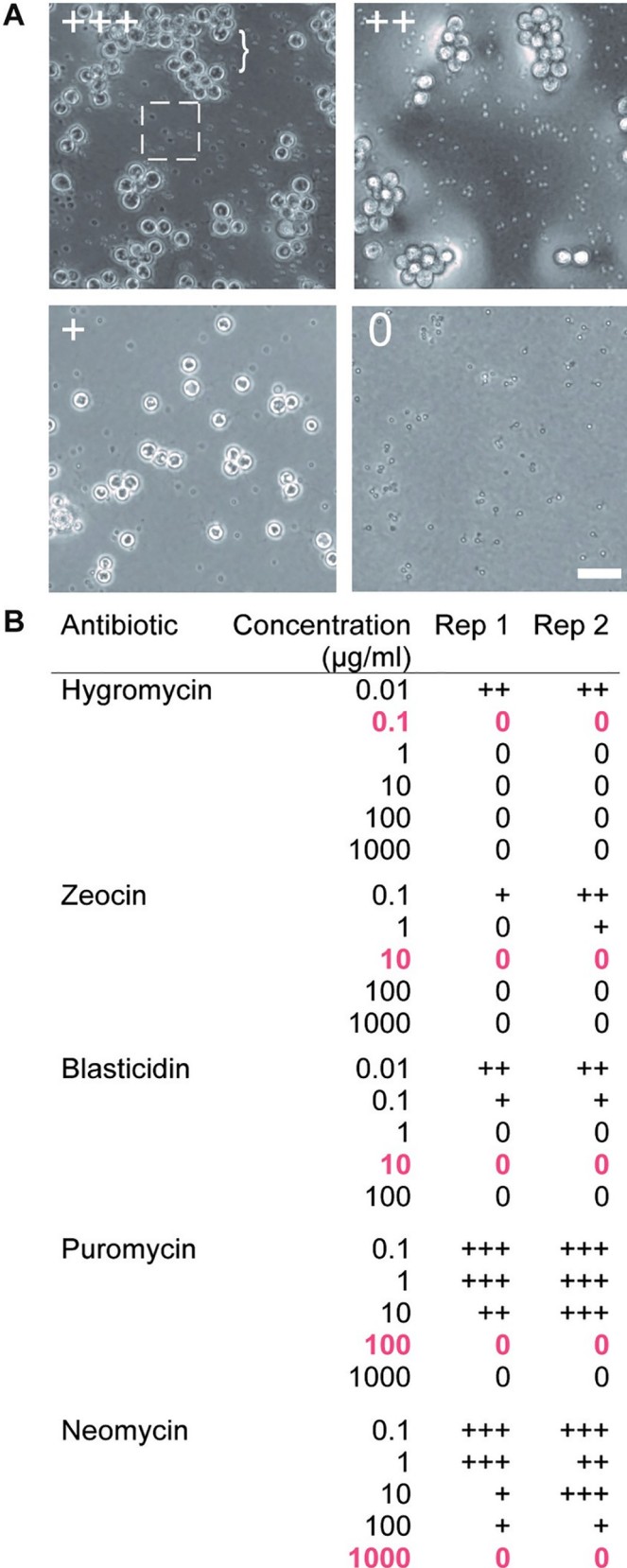

| Antibiotic | Concentration (µg/ml) | Rep 1 | Rep 2 |
|---|---|---|---|
| Hygromycin | 0.01 | ++ | ++ |
| | **0.1** | **0** | **0** |
| | 1 | 0 | 0 |
| | 10 | 0 | 0 |
| | 100 | 0 | 0 |
| | 1000 | 0 | 0 |
| Zeocin | 0.1 | + | ++ |
| | 1 | 0 | + |
| | **10** | **0** | **0** |
| | 100 | 0 | 0 |
| | 1000 | 0 | 0 |
| Blasticidin | 0.01 | ++ | ++ |
| | 0.1 | + | + |
| | 1 | 0 | 0 |
| | **10** | **0** | **0** |
| | 100 | 0 | 0 |
| Puromycin | 0.1 | +++ | +++ |
| | 1 | +++ | +++ |
| | 10 | ++ | +++ |
| | **100** | **0** | **0** |
| | 1000 | 0 | 0 |
| Neomycin | 0.1 | +++ | +++ |
| | 1 | +++ | ++ |
| | 10 | + | +++ |
| | 100 | + | + |
| | **1000** | **0** | **0** |

**Fig 4. Inhibition of Bd growth on solid media.** (**A**) Examples of *Bd* growth after three days on antibiotic selection plates. The '+' demonstrates the relative zoospore activity of each plate compared to an antibiotic-free control plate. The box highlights zoospores, which appear as small dots while the bracket highlights sporangia. The zoospores in the '0' image are immotile (see S1 Video). Scale bar 50 μm. (**B**) *Bd* growth on antibiotic selection plates. Concentrations highlighted in bold and orange are the lowest concentrations that prevent growth for at least 14 days post zoospore plating.

media, the solid media typically required a higher concentration of antibiotic. This may be because of the additional minerals found in the agar not present in the liquid media [66]. Hygromycin, Zeocin, and Blasticidin worked well for both species and at concentrations within the typical range used for genetic selection in other species (**Table 1**). Puromycin and Neomycin were both able to inhibit growth of *Bd* and *Bsal*, but required higher concentrations than are used for animal cell lines. Although Hygromycin, Zeocin, and Blasticidin are all effective for preventing growth of *Bd* and *Bsal*, we recommend first using Hygromycin for genetic selection because it has been successfully used for selection of transformants in the nonpathogenic chytrid *Spizellomyces punctatus*, and is widely used for other fungal species [8, 36–38].

The ability to select for genetically transformed cells will allow for tractable genetic models to facilitate hypothesis testing in *Bd* and *Bsal*. The identification of useful selection agents and appropriate working concentrations is an important first step in developing genetic tools for use with *Bd* and *Bsal*. The natural step forward will be the design of selection cassettes, most commonly in the form of transformation plasmids. We look forward to the development of these and related molecular tools that will help us answer questions about the basic cell biology of chytrids, fungal evolution, and amphibian pathology.

## Methods

### Cell growth and synchronization

*Batrachochytrium dendrobatidis* (*Bd*) isolate JEL 423 was grown in 1% (w/v) tryptone (Apex Cat. 20–251) in tissue culture treated flasks (Cell Treat 229340) at 24˚C for three days. *B. salamandrivorans* (*Bsal*) isolate AMFP 1 was grown in half-strength TGhL liquid media (0.8% Tryptone, 0.2% gelatin hydrolysate, 0.1% lactose (w/v) in tissue culture treated flasks at 15˚C for four days [67]. For both species, we synchronized the release of motile zoospores by gently washing the flask three times with fresh growth media and then incubating with 10 mL of media for 2 hours. Age matched zoospores were then collected by centrifugation at 2000 rcf for 5 mins, resuspended in media, counted, and used for experiments as outlined below.

### Drug treatments and quantitation for cells grown in liquid media

Neomycin (Fisher Cat. AAJ67011AE), Hygromycin B (Fisher Cat. AAJ60681MC), Blasticidin (Fisher Cat. BP2647100), Puromycin (Fisher Cat. BP2956100), and Zeocin (Fisher Cat. AAJ671408EQ), were screened for growth inhibition of *Bd* and *Bsal*. Cells were diluted to a starting concentration of $5 \times 10^5$ cells/mL and 250 uL of cells were added to each well of a sterile tissue culture treated 24-well plate (Cell Treat 229123). 250 μl of appropriately diluted antibiotics and matched carrier controls were added to each well and mixed thoroughly. Plates were sealed with parafilm and grown at either 24˚C for three days (*Bd*), or 15˚C for four days (*Bsal*). For each of three biological replicates spaced in time, the concentration of released zoospores was estimated using the average of two independent hemocytometer counts. Zoospore concentrations were normalized to the no drug control and data plotted using Prism (GraphPad v8).

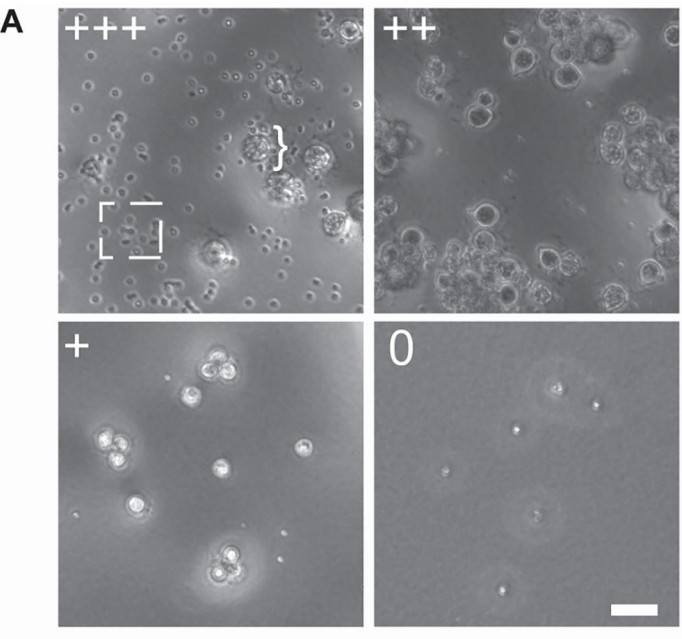

| Antibiotic | Concentration µg/ml | Rep 1 | Rep 2 |
|---|---|---|---|
| Hygromycin | 0.001 | +++ | +++ |
|  | 0.01 | +++ | ++ |
|  | 0.1 | ++ | + |
|  | 1 | 0 | 0 |
|  | **10** | **0** | **0** |
| Zeocin | 0.01 | ++ | ++ |
|  | 0.1 | 0 | 0 |
|  | 1 | 0 | 0 |
|  | **10** | **0** | **0** |
|  | 100 | 0 | 0 |
| Blasticidin | 0.01 | + | ++ |
|  | 0.1 | + | 0 |
|  | 1 | 0 | 0 |
|  | **10** | **0** | **0** |
|  | 100 | 0 | 0 |
| Puromycin | 0.1 | +++ | +++ |
|  | 1 | ++ | +++ |
|  | 10 | ++ | + |
|  | 100 | + | 0 |
|  | 500 | 0 | 0 |
| Neomycin | 0.1 | ++ | ++ |
|  | 1 | ++ | ++ |
|  | 10 | ++ | +++ |
|  | 100 | + | + |
|  | 1000 | 0 | 0 |

**Fig 5. Inhibition of *Bsal* growth on solid media.** (**A**) Examples of *Bsal* growth after four days on antibiotic selection plates. The '+' demonstrates the relative zoospore activity of each plate compared to a no antibiotic control plate. The box highlights zoospores, which appear as small dots while the bracket highlights sporangia. The zoospores in the '0' image are immotile (see S1 Video). Scale bar 50 μm. (**B**) *Bsal* growth on antibiotic selection plates. Concentrations highlighted in bold and orange are the lowest concentrations that prevent growth for at least 14 days post zoospore plating.

## Drug treatments and quantitation for cells grown on solid media

We added 1% agar to 50 mL batches of 1% tryptone (w/v) and half-strength TGhL then auto-claved. Each antibiotic was added to a separate, pre-cooled, 50 mL batch of media, and 10 mL of the solution added to one of five 15 mm$^2$ plates (VWR 25384–090) and allowed to solidify. Equal volume of appropriate carrier liquid was added to the pre-cooled 50 mL batch of agar-media to create control plates. Plates were wrapped in parafilm and aluminum foil, and stored at 4°C. Plates were inoculated by evenly spreading 5.0 x 10$^6$ zoospores across the agar and incubated at 24°C for three days (*Bd)* or 15°C for four days (*Bsal*). Three control plates were used per replicate to ensure a point of comparison if one were to be contaminated. Zoospore release was evaluated by imaging each plate for 20 seconds at one second intervals using a Nikon Ti2-E inverted microscope equipped with 10x PlanApo objective and sCMOS 4mp camera (PCO Panda) using white LED transmitted light. Approximate zoospore activity was assessed as: 0 (no visible zoospores), + ($<$ 25% zoospore activity of control plates lacking anti-biotic), ++ (~50% zoospore activity of control plates), or +++ (equivalent zoospore activity to control plates). To determine the lowest antibiotic concentration that could completely inhibit growth, plates that yielded "0" growth were allowed to grow for 14 days at the appropriate incubation temperature and reassessed as above.

## Supporting information

**S1 Video. *Bsal* zoospores with zero growth.** Zoospores grown on antibiotic selection plates are labeled "0" if no zoospores are released or zoospores showed no growth and are immotile. (MP4)

**S2 Video. *Bsal* zoospores with "+++" growth.** Zoospores grown on antibiotic selection plates are labeled "+++" if the zoospore release is comparable to the no antibiotic control. (MP4)

## Author Contributions

**Conceptualization:** Lillian K. Fritz-Laylin.

**Data curation:** Kristyn A. Robinson.

**Formal analysis:** Kristyn A. Robinson, Mallory Dunn.

**Funding acquisition:** Lillian K. Fritz-Laylin.

**Investigation:** Kristyn A. Robinson, Mallory Dunn, Shane P. Hussey.

**Methodology:** Mallory Dunn, Shane P. Hussey.

**Project administration:** Lillian K. Fritz-Laylin.

**Supervision:** Lillian K. Fritz-Laylin.

**Validation:** Kristyn A. Robinson.

**Visualization:** Kristyn A. Robinson.

**Writing – original draft:** Kristyn A. Robinson, Lillian K. Fritz-Laylin.

**Writing – review & editing:** Kristyn A. Robinson, Mallory Dunn, Shane P. Hussey, Lillian K. Fritz-Laylin.

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
