## [Decision Letter · Decision Letter 0]

14 Aug 2020

PONE-D-20-22012

Identification of antibiotics for use in selection of the chytrid fungi Batrachochytrium dendrobatidis and Batrachochytrium salamandrivorans

PLOS ONE

Dear Dr. Dr. Fritz-Laylin:

Thank you for submitting your manuscript to PLOS ONE. After careful consideration, we feel that it has merit but needs some minor revisions to address reviewer comments. Therefore, we invite you to submit a revised version of the manuscript that addresses the points raised during the review process.

Your manuscript has been reviewed by two experts in fungal biology and chytridiomycosis. Both had favorable comments about the manuscript.  Thus, my recommendation is for minor revisions.  Please revise and address the comments of both reviewers in a point-by-point response.

We look forward to receiving your revised manuscript.

Kind regards,

Louise A. Rollins-Smith

Academic Editor

PLOS ONE

Journal Requirements:

Reviewers' comments:

Reviewer's Responses to Questions

**Comments to the Author**

1. Is the manuscript technically sound, and do the data support the conclusions?

Reviewer #1: Yes

Reviewer #2: Yes

2. Has the statistical analysis been performed appropriately and rigorously? 

Reviewer #1: N/A

Reviewer #2: N/A

3. Have the authors made all data underlying the findings in their manuscript fully available?

Reviewer #1: Yes

Reviewer #2: Yes

4. Is the manuscript presented in an intelligible fashion and written in standard English?

Reviewer #1: Yes

Reviewer #2: Yes

5. Review Comments to the Author

Reviewer #1: This manuscripts provides a systematic examination of the drug sensitivity of the amphibian pathogenic fungi Batrachochytrium which is a critical step in development of tools for transformation.

This is a clearly written manuscript with clearly presented results and a straightforward summary. I do not have a lot of criticisms to raise.

While the costs for reagents are helpful to compare in the table, the dosage used is also quite variable. If there could be normalization of costs for the dosage needed that would be the most fair comparison of relative costs I think.

It would be helpful if a protocols.io protocol also accompanied this publication for additional description of methods and reagents.

Video supplemental files are great addition as well to demonstrate motility and growth.

Figure 1 is a little dark to read but overall this is a minor issue.

Reviewer #2: Robinson et al. determined viable antibiotic concentrations to inhibit the growth of Bd and Bsal in liquid and solid media. The goal of finding these concentrations was to pave the way for future genetic manipulation of these species by providing a reliable way to screen for transformants using antibiotic resistance. The authors tested a range of concentrations for 5 different antibiotics that are commonly used in mammalian and fungal species. They were able to determine usable concentrations for each antibiotic in both liquid and solid media that could completely inhibit motile zoospore production and growth. It is useful that the authors waited until 14 days of potential growth on solid media to make firm decisions.

This was important result that displays some fundamental work that must be done in order to better understand the genetics and pathology of Bd and Bsal which have decimated amphibian populations worldwide. Once work on genetic manipulation of these species is started, having a way to screen for successful transformants will be essential to expedite the process. Robinson et al. are laying the groundwork for future experiments and contributing to the creation of methods for manipulating chytrid species.

Most of the comments are suggestions to improve clarity.

The authors did not mention how the no drug control was created. Was media/water added instead of drug? Were the samples run in same plate as experimental samples? Were there multiple no drug control replicates? The no drug controls on solid media should be clarified also.

Line 54: what about chytrids lends to them having a lack of genetic tools? A better introduction as to what the difficulties have been would be useful to contextualize. You went through all the trouble of explaining chytrid lifecycle, is this relevant to these problems? Also, what genetic tools are on the horizon? I think some specific examples could be useful (like CRISPR).

Should the authors discuss why they settled on their approach at counting zoospores rather than using optical density and measuring over a longer period of time? I think ultimately the data seem robust, but there is a large error in estimating zoospore number using a hemocytometer. Were multiple measurements for each replicate well?

Table 1. I find it hard to read. Would it make sense to transpose it so the rows are drugs and columns variables?

Line 33: I would say rather, “Encysted spores of many species develop directly into sporangia and develop…”

Figure 1 is pretty dark. I suggest somehow increasing brightness, though it may just be in the copy that was transmitted.

Figure 2: some low concentrations grew better than control? Please explain. Is this due to measurement error or the controls being prepared in a different way?

Figures 2 & 3, please add legend directly to figure.

6. PLOS authors have the option to publish the peer review history of their article (what does this mean?). If published, this will include your full peer review and any attached files.

Reviewer #1: No

Reviewer #2: No

---

## [Author Response · Author response to Decision Letter 0]

24 Sep 2020

Reviewer #1: 

This manuscripts provides a systematic examination of the drug sensitivity of the amphibian pathogenic fungi Batrachochytrium which is a critical step in development of tools for transformation.

This is a clearly written manuscript with clearly presented results and a straightforward summary. I do not have a lot of criticisms to raise.

While the costs for reagents are helpful to compare in the table, the dosage used is also quite variable. If there could be normalization of costs for the dosage needed that would be the most fair comparison of relative costs I think.

Thank you for pointing this out. We have added two columns to the table for “costs per liter” for Bd and Bsal. The costs are based off of the lowest concentrations required to inhibit chytrid growth in both liquid and solid media.

It would be helpful if a protocols.io protocol also accompanied this publication for additional description of methods and reagents.

Thank you for this suggestion, we are currently assembling a step-by-step protocol for deposition to protocols.io and will make this publicly available as soon as possible 

Video supplemental files are great addition as well to demonstrate motility and growth.

We’re glad you enjoyed the supplemental videos and found them useful.

Figure 1 is a little dark to read but overall this is a minor issue.

We have adjusted the brightness of the images for clarity.

Reviewer #2: 

Robinson et al. determined viable antibiotic concentrations to inhibit the growth of Bd and Bsal in liquid and solid media. The goal of finding these concentrations was to pave the way for future genetic manipulation of these species by providing a reliable way to screen for transformants using antibiotic resistance. The authors tested a range of concentrations for 5 different antibiotics that are commonly used in mammalian and fungal species. They were able to determine usable concentrations for each antibiotic in both liquid and solid media that could completely inhibit motile zoospore production and growth. It is useful that the authors waited until 14 days of potential growth on solid media to make firm decisions.

This was important result that displays some fundamental work that must be done in order to better understand the genetics and pathology of Bd and Bsal which have decimated amphibian populations worldwide. Once work on genetic manipulation of these species is started, having a way to screen for successful transformants will be essential to expedite the process. Robinson et al. are laying the groundwork for future experiments and contributing to the creation of methods for manipulating chytrid species.

Most of the comments are suggestions to improve clarity.

The authors did not mention how the no drug control was created. Was media/water added instead of drug? Were the samples run in same plate as experimental samples? Were there multiple no drug control replicates? The no drug controls on solid media should be clarified also.

Clarifications on the no drug control have been added to the text as follows:

For liquid growth experiments (lines 231-234): “250 µl of appropriately diluted antibiotics and matched carrier controls were added to each well and mixed thoroughly. Plates were sealed with parafilm and grown at either 24 °C for three days (Bd), or 15 °C for four days (Bsal). For each of three biological replicates spaced in time, the concentration of released zoospores was estimated using the average of two independent hemocytometer counts.”

For plate growth experiments (lines 242-246): “Equal volume of appropriate carrier liquid was added to the pre-cooled 50 mL batch of agar-media to create control plates. Plates were wrapped in parafilm and aluminum foil and stored at 4 °C. Plates were inoculated by evenly spreading 5.0 x 106 zoospores across the agar and incubated at 24 °C for three days (Bd) or 15 °C for four days (Bsal). Three control plates were used per replicate to ensure a point of comparison if one were to be contaminated.”

Line 54: what about chytrids lends to them having a lack of genetic tools? A better introduction as to what the difficulties have been would be useful to contextualize. You went through all the trouble of explaining chytrid lifecycle, is this relevant to these problems? Also, what genetic tools are on the horizon? I think some specific examples could be useful (like CRISPR).

The biggest hurdle to genetic tools for chytrids, in our opinion, is simply the relatively few labs that are making systematic attempts at developing new tools. This is, however, rapidly changing and tools are currently being developed. We have made changes to the text and the section now reads as follows:

“However, studying the molecular mechanisms driving pathogenesis remains challenging due to the lack of genetic tools available for chytrid fungi. Electroporation protocols have been developed for Bd and Bsal, which could be used to deliver molecular payloads for genetics manipulation such as plasmids and/or CRISPR-Cas9 complexes (12). The recent success in genetic manipulation of a related chytrid species, Spizellomyces punctatus (Sp), is a major breakthrough for our ability to study chytrid biology (8). We and others are now striving to adapt this technology to Bd and Bsal to further our understanding of chytridiomycosis.”

Should the authors discuss why they settled on their approach at counting zoospores rather than using optical density and measuring over a longer period of time? I think ultimately the data seem robust, but there is a large error in estimating zoospore number using a hemocytometer. Were multiple measurements for each replicate well?

We decided to use a hemocytometer because it allows us to differentiate between live and dead cells, unlike measuring ODs, which cannot differentiate between live and dead cells. Because the data was consistent across three independent biological trials, we are confident that the concentrations we report are based on robust data. We did not repeat these measurements using a plate reader, but would expect to see similar results based on the plate growth assays.

Table 1. I find it hard to read. Would it make sense to transpose it so the rows are drugs and columns variables?

Thanks, we have taken this suggestion.

Line 33: I would say rather, “Encysted spores of many species develop directly into sporangia and develop…”

We have edited the text to read:

“Encysted spores of many species develop into sporangia and develop hyphal-like structures called rhizoids and grow rapidly.”

Figure 1 is pretty dark. I suggest somehow increasing brightness, though it may just be in the copy that was transmitted.

We have adjusted the brightness of these images.

Figure 2: some low concentrations grew better than control? Please explain. Is this due to measurement error or the controls being prepared in a different way?

None of the low concentration samples gave higher-than-control growth for all replicates. Additional measurements of cells on the same day or measuring the same sample over longer periods of time does not resolve this inherent variability, so we see this as a biological variability rather than measurement error. This is why we conduct multiple experimental replicates conducted on different days. 

Figures 2 & 3, please add legend directly to figure.

Thanks for this suggestion. We are following PLOS ONE’s requirements for submission of revised manuscripts for this submission.

---

## [Editor Report · Decision Letter 1]

28 Sep 2020

Identification of antibiotics for use in selection of the chytrid fungi Batrachochytrium dendrobatidis and Batrachochytrium salamandrivorans

PONE-D-20-22012R1

Dear Dr. Fritz-Laylin

We’re pleased to inform you that your manuscript has been judged scientifically suitable for publication and will be formally accepted for publication once it meets all outstanding technical requirements.

Kind regards,

Louise A. Rollins-Smith

Academic Editor

PLOS ONE

Additional Editor Comments (optional):

Thank you for submitting your revised manuscript.  I have looked at the responses to reviewers and the manuscript with track changes.  It is my view that you have addressed the reviewer concerns, and the manuscript is accepted.
---

## [Editor Report · Acceptance letter]

2 Oct 2020

PONE-D-20-22012R1 

Identification of antibiotics for use in selection of the chytrid fungi *Batrachochytrium dendrobatidis and Batrachochytrium salamandrivorans*

Dear Dr. Fritz-Laylin:

I'm pleased to inform you that your manuscript has been deemed suitable for publication in PLOS ONE. Congratulations! Your manuscript is now with our production department. 

Kind regards, 

on behalf of

Dr. Louise A. Rollins-Smith 

Academic Editor

PLOS ONE